tumour mutation burden; immune checkpoint inhibitors; whole exome sequencing; targeted gene panels

**Corresponding author:**
Attia M. Elbehi;
Email: mattia@alumni.harvard.edu

# The challenges and opportunities of applying tumour mutational burden analysis to precision cancer medicine

Attia M. Elbehi ⓘD

Department of Oncology, Medical Sciences Division, University of Oxford, Oxford, UK

## Abstract

The discovery and development of immune checkpoint inhibitors (ICIs) has revolutionised the management of human cancers. However, only a subset of patients responds to ICI therapy, even though immune evasion is a hallmark of cancer. Initially, treatment was administered to patients on the basis of expression levels of one of the targets of ICI therapy, programmed cell death ligand 1. In clinical trials, the high response rate of melanoma and non-small cell lung cancer patients to ICI therapy supported the basic premise of cancer immunotherapy, that tumour-specific mutated proteins trigger an immune response. Tumour mutational burden subsequently emerged as a potential biomarker for response to ICI therapy. This review summarises the evidence supporting the scientific rationale for TMB as a biomarker for ICI therapy and focuses on some of the major challenges associated with incorporation of TMB into routine clinical practice.

## Impact statement

The tumour mutation burden (TMB) has emerged as a promising predictive biomarker for cancer immunotherapy. This review aims to provide a comprehensive and in-depth examination of the different methods used to quantify TMB and their associated limitations and challenges. This study explored potential solutions to improve the standardisation and accuracy of TMB assessment. This thorough examination may advance the field of precision cancer medicine and improve patient outcomes.

## Introduction

Cancer is a global health issue and the second leading cause of death worldwide. The GLOBO-CAN reported high cancer incidence with 19.3 million new cases and 10 million mortalities in the year of 2020 (Sung et al. 2021). Cancer is a genetic disease, and as described by Hanahan and Weinberg, one of the hallmarks of cancer is genomic instability and mutations (Hanahan and Weinberg 2011). Somatic mutations in human cancers have been central to the design of methods to distinguish cancer cells from normal cells. The discovery that the average adult solid tumour may harbour ~90 amino acid-altering somatic mutations has led to further appreciation of these mostly nonsynonymous mutations for their potential to produce non-self antigens acting as a trigger for the host's own adaptive immune response (Segal et al. 2008). Hanahan and Weinberg also reported that one of the characteristic features of cancer is the development of immune evasion strategies; and therefore, the concept of utilising the immune system to attack and eliminate cancer cells has been speculated for a long time; however, the precise underlying tumour escape mechanisms were poorly understood until very recently (Hanahan and Weinberg 2011). Over the past decade, diverse translational research has been conducted to develop a better understanding of the tumour immunobiology. Consequently, James Allison and Tasuku Honjo were awarded the Nobel Prize in Physiology or Medicine for the discovery of immune checkpoints CTLA-4 and PD-1, which are inhibitory proteins produced or secreted by cancer cells to suppress and evade T-cell recognition and immune system activation. In addition, several inhibitory immune checkpoints such as CTLA-4, PD-1, LAG-3, TIM-3, and TIGIT have been identified as therapeutic targets for immunotherapy. Of these, CTLA-4 and PD-1 have been most extensively studied immune checkpoint inhibitors (ICIs), and the U.S. Food and Drug Administration (FDA) has approved several monoclonal antibodies targeting both pathways (Greenwald et al. 2005; Parry et al. 2005; Dougan and Dranoff 2009; Sakuishi et al. 2010; Mellman et al. 2011; OPDIVO 2018; TECENTRIQ 2019; BAVENCIO 2020; IMFINZI 2020; KEYTRUDA 2021). The manipulation of the immune system with immune checkpoint inhibitors (ICIs) which relieve immune blockade in human tumours, has fulfilled the potential of these cancer-specific antigens and brought about a new era in cancer treatment of a potentially agnostic approach to

cancer therapy. However, not all patients respond. Thus, the research efforts have been devoted to identifying biomarkers that distinguish responsive tumours from non-responsive tumours.

Historically, several studies have highlighted the immunogenic nature of melanoma, as demonstrated by spontaneous tumour regression, and the remarkably durable benefits of Interleukin-2 therapy in a small subset of patients that is lasting for over 10 years. This may be attributed to the excessive exposure of melanocytes to ultraviolet radiation, and therefore the accumulation of a higher number of mutations than in other cancers. Similarly, for lung cancer, and although it was not initially considered an immune-responsive tumour, ICIs have demonstrated substantial survival improvement in patients with non-small cell lung cancer (NSCLC) (Payne et al. 2014; Ong et al. 2016). Consequently, the association between high mutational load and the favourable immunotherapy response in melanoma and NSCLC has led to the emergence of the tumour mutation burden (TMB) as a potential biomarker.

### Is TMB an accurate predictor of ICI response?

TMB is rigorously defined as the total number of somatic mutations within the tumour genome; however, in practice it involves an estimate from a subset of the genome. The efficiency of ICIs is based primarily on the ability of the immune system, predominantly the T-cells, to recognise and attack cancerous cells. The T-cell activation could be triggered by cancer antigen recognition. The accumulation of somatic alterations in DNA may lead to neoplastic transformation and cancer cell development. These include synonymous mutations (silent mutations that do not alter amino acid coding), non-synonymous mutations (largely comprised of nonsense and point mutations that change the amino acid codon), insertions or deletions (indels, which can cause frameshifts), copy number variants (CNVs), and gene fusions. However, not all somatic mutations generate foreign or non-self antigens, known as neoantigens, which can be recognised by the immune system and are able to elicit immune reaction. For immune system activation, such mutations need to be transcribed and translated into specific neoantigens that could be caught up by the APCs and bound to MHC molecules for further presentation on the cell surface. Furthermore, a higher TMB corresponds to a higher number of somatic mutations and high neoantigen load. Thus, there is an increasing probability that these neoantigens could be recognised by cytotoxic T-cells and elicit an immunogenic response, leading to the destruction of cancer cells, as illustrated in Figure 1 (Garcia-Lora et al. 2003; Chen and Mellman 2013; Wirth and Kühnel 2017;

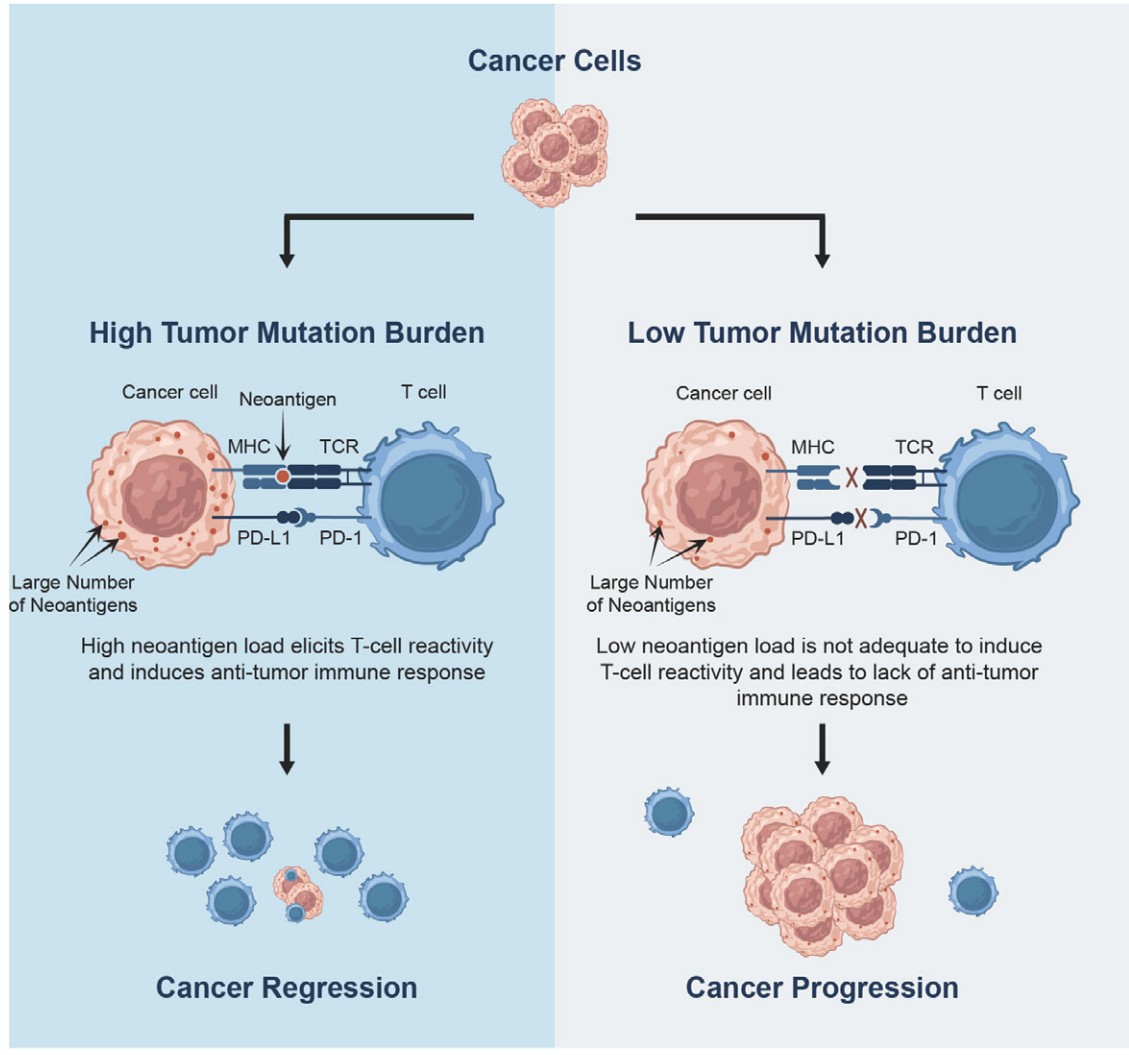

**Figure 1.** The association between Tumor Mutation Burden and responses to cancer immunotherapy.

Lang et al. 2022). Therefore, TMB has been extensively studied in lung cancer and validated as an independent predictive biomarker (Wirth and Kühnel 2017).

In the KEYNOTE-158 (Marabelle et al. 2020) is an open-label, multi-cohort trial of pembrolizumab in patients with advanced multiple cancer types that progressed despite prior therapies and had no satisfactory treatment options. The study utilised the FoundationOne CDx (F1CDx) assay for TMB estimation and the cut-off for TMB-H was ≥10 and ≥ 13 mut/Mb. The trial included 1,050 patients in total and 790 were evaluated for TMB assessment. A total of 102 patients (13%) belonged to the TMB-H group. The study reported an ORR of 29.4% in patients with TMB-H, of whom 3.9% and 25.4% showed complete and partial responses, respectively, versus an ORR of 6.3% in patients with TMB < 10 muts/Mb. The median duration of response (DOR) was not reached in the TMB-H group; however, it was > 2 years in two-thirds (66.6%) of the responders. Interestingly, ORR was only 13% in patients with TMB ≥ 10 mut/Mb and < 13 mut/Mb compared with 37% in those with ≥13 mut/Mb. A retrospective analysis for TMB using WES from 12 trials investigated pembrolizumab monotherapy (KEYNOTE-001, 002, -010, -012, -028, -045, -055, -059, -061, -086, -100, and -199). TMB was assessed as the number of non-synonymous SNVs and indels found in protein-coding regions and TMB-H was defined as ≥175 mut/exome. A total of 2,234 patients were evaluated for WES TMB results (1,772 received pembrolizumab and 462 received chemotherapy), and approximately 24% belonged to TMB-H category. In concordance with the KEYNOTE-158 results, patients with TMB-H (≥ 175 mut/exome) showed a higher ORR of 31.4% compared with that of 9.5% in patients with TMB-L (< 175 mut/exome). Based on these results, the US FDA granted an accelerated approval to pembrolizumab for the treatment of adult and paediatric patients with unresectable or metastatic TMB-H (≥ 10 mut/Mb) solid tumours that progressed after prior treatment and had no satisfactory alternative treatment options (Cristescu et al. 2020; Marabelle et al. 2020; Pembrolizumab prescribing information 2020).

CheckMate 568 (Ready et al. 2019) is a single-arm, open-label, phase II trial study investigated the association of TMB with response to nivolumab plus ipilimumab in NSCLC. The study reported that median progression-free survival (PFS) was longer in patients with TMB-H (7.1 months [95% CI, 3.6–11.3 months]) versus TMB-L (2.6 months [95% CI, 1.4 to 5.4 months]), with PFS rate of 55% and 31% at 6 months for the TMB-H and TMB-L subgroups, respectively. Thus, CheckMate 568 has validated the predictive ability of TMB as an independent biomarker of response to nivolumab plus ipilimumab treatment in NSCLC, irrespective of the tumour PD-L1 expression level, and also provided important insights on the TMB threshold (Ready et al. 2019). However, the reliance on TMB is not as feasible as it appears since TMB is associated with several challenges or remaining questions to personalised treatment of cancer patients. First, what methods should be used to accurately and cost-effectively determine TMB in clinical practice? Second, what are the threshold levels of TMB high in various tumour types? In this review, we discuss the methods for the determination of TMB in tumours and the subsequent challenges.

## TMB challenges and special consideration

There are various issues that impact the accurate quantification of TMB and hinder its broad utilisation in the clinic, as summarised in Figure 2.

## TMB measurement, validation and pre-analytical considerations

In general, the incorporation of new cancer biomarkers, particularly those that need enough tissue, into routine clinical practice is very demanding since it should be backed up with strong clinical evidence. In addition, the test should be performed with a minimal amount of DNA, have a reasonable cost to be reimbursed and turnaround time that do not significantly delay therapeutic interventions, and provide accurate results. This is even more challenging with TMB, owing to its complex NGS workflow and the need for in-depth bioinformatics expertise. TMB estimation needs larger amount of high-quality DNA than those for single gene testing, WGS requires between 50 nanograms and 1 microgram of high-quality DNA and therefore it is critical to obtain enough tissues to overcome this issue and address tumour heterogeneity and avoid false-negative results. It is not only about quantity but also the quality of the DNA is even more important. Moreover, there should be an adequate percentage of viable tumour nuclei within the sample. For a single-gene testing tools such as Sanger sequencing, 40% of tumour DNA is enough for the detection of variants; however, for WGS which includes broader and more comprehensive coverage, so a larger genetic content is required. Therefore, The Cancer Genome Atlas (TCGA) excludes tissues with 20–50% necrosis and necessitates samples with greater, 60–70%, tumour nuclei, this criterion is even stricter for glioblastoma multiforme (GBM) and requires 80% tumour nuclei. One solution is to improve sample quality by dissecting and removing necrotic areas before analysis (The Cancer Genome Atlas Research Network 2008). The current process for DNA fixation is the formalin-fixed, paraffin-embedded (FFPE), which is associated with many drawbacks, has that can lead to DNA damage. Instead, recent studies have considered Fresh Frozen (FF) for tissue fixation and preservation to overcome formalin damage. Although FF has also several issues but primarily logistical related to storage at ultralow temperature, using liquid nitrogen (LN), which is extremely expensive, and such infrastructure is not widely available in hospitals. Moreover, there is risk of sample damage in case of temperature changes and also serious risks, such burns, tank explosions, and suffocation in case of LN2 leakage. Most importantly, FF provides high quality DNA compared to FFPE (FFPE vs Frozen Tissue Samples 2018; Fresh vs Frozen Samples: Human Clinical Samples 2018; Robbe et al. 2018).

Another challenge in tissue sample-based assays is the tumour heterogeneity which refers to the presence of genetic and phenotypic discrepancies within a tumour or between different regions of the same tumour which can impact TMB estimation in several ways. Subclonal mutations: Tumours often contain subpopulations of cells with different genetic profiles, where some mutations may be present in only a small fraction of tumour cells. This can lead to an underestimation of TMB if the assay does not capture all the subclonal mutations. Second, spatial heterogeneity: different regions of a tumour may have distinct mutation profiles; therefore, the biopsy of a single region may not capture the most mutated region, leading to an inaccurate estimation of TMB. Third, temporal heterogeneity: Tumours can evolve over time, acquiring new mutations or losing existing ones; thus, a single biopsy may not capture the full spectrum of mutations present at different stages of tumour development. This can lead to variability in TMB estimation if relying on an archived tissue that does not align with the most recent mutational load (Schmelz et al. 2021).

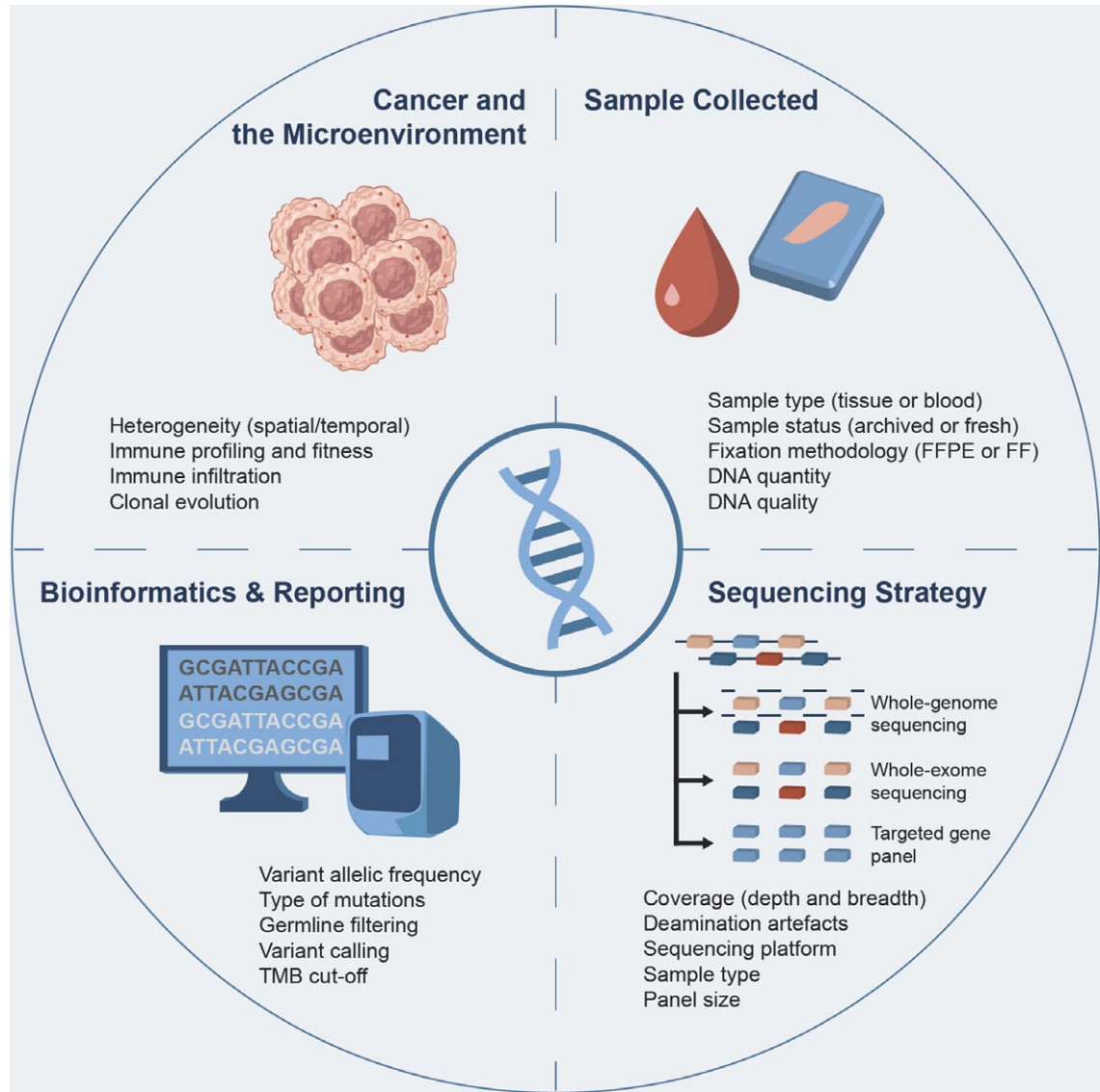

**Figure 2.** Factors affecting the estimation and reporting of Tumor Mutation Burden.

In recent years, the analysis of circulating tumour DNA (ctDNA), commonly referred to as liquid biopsy, has undergone substantial advancements. This methodology possesses significant potential to address numerous challenges previously outlined. The sequencing of ctDNA yields critical insights into the dynamics of the oncogenic mutational landscape. Furthermore, it serves as a real-time biomarker that facilitates the accurate and timely assessment of TMB. Additionally, liquid biopsy offers a noninvasive tool for the continuous monitoring of therapeutic responses, evaluation of minimal residual disease, and early detection of disease progression indicators (Sivapalan et al. 2023).

### Variation in breadth and depth of coverage

The genome coverage varies according to the assay or platform. Whole exome sequencing (WES) covers only the protein coding regions, accounting for approximately 1–2% of the human genome, and around 22,000 genes with 30–50 Mb in size. Thus, WES has the capacity to detect most of the genetic variants associated with diseases. In contrast, targeted gene panels cover a smaller range of size and number of genes, for example, FoundationOne CDx covers a total 0.8 Mb and 324 genes, while the MSK-IMPACT assay covers a total of 1.5 Mb and 468 genes. Clinical studies have indicated that gene panels smaller than these may be insufficient for accurate TMB estimation. Inconsistent TMB measurements have been associated with panels covering <0.5 Mb of the genome. Gene panels of ≥0.8 Mb are therefore essential for the accurate TMB estimation (Ng et al. 2009; Baras et al. 2017; Evaluation of Automatic Class III Designation for MSK-IMPACT (Integrated Mutation Profiling of Actionable Cancer Targets): decision summary 2018; FDA 2018b). The depth of sequencing is also important and it too varies significantly based on the various NGS assays or the platforms used. The minimum coverage depth required for precise TMB estimation is around 200×. However, WES provides ~100×, and can only detect mutations with allele frequencies >15%. In contrast, gene panels provide deeper coverage at approximately

500×, which improves the detection of low-frequency variants. Therefore, gene panels can provide adequate coverage and reliable TMB estimation (Cheng et al. 2015; Feliubadaló et al. 2017; Lee et al. 2017).

## Variation in TMB estimation

The TMB estimation varies based on multiple factors, including the NGS platforms, panel size, depth of coverage, somatic variants/mutations counted, and TMB threshold. In the meantime, a standardised method for TMB analysis, interpretation, and result reporting remains undetermined. A recent study by the Quality in Pathology (QuIP) reported that up to 25% of samples had been misclassified as TMB-H and TMB-L. The laboratories included in this study utilised various TMB methods, including commercially available techniques such as Oncomine™, while other centres developed their own panels for TMB estimation. Moreover, the type of mutations considered for TMB detection and cutoff TMB values used for result interpretation also varied significantly between the participating laboratories. Collectively, such discrepancies led to inconsistent interpretations of the results, negatively impacted the clinical utility, and limited the widespread utilisation of TMB as a predictive biomarker. Furthermore, 19 laboratories used cell-free DNA (cfDNA) to quantify TMB, despite of

the limited evidence on its sensitivity and specificity for TMB testing, as well as the very low allelic frequency of variants that could be detected in the peripheral blood (Fenizia et al. 2018; Gandara et al. 2018; Stenzinger et al. 2020). These findings raise serious concerns on the reproducibility of TMB results and reinforce the urgent need for standardisation, validation, and clinical accreditation of TMB. Additionally, the Friends of Cancer Research (FoCR) TMB Harmonisation Project study has reported that filtering out the pathogenic variants is critical to avoid the overestimation of TMB. Table 1 summarises the various types of the available TMB assays.

## Differences in NGS approaches or platforms

There are different workflows that can be used for TMB analysis: WGS, WES, or large targeted gene panels, and each has its advantages and disadvantages. The WGS Workflow provides the most comprehensive since it covers the entire genome. Thus, it can detect almost all types of genetic variants which lead to the most accurate estimation of TMB. However, it requires the highest sequencing depth and coverage and generates large amounts of data, requiring more computational resources for analysis, subsequently it is the most expensive and resource-intensive workflow. WES Workflow is regarded as the gold standard

**Table 1.** Summary of the various available assays and platforms for TMB estimation

| TMB assay | No. of genes and Mbs covered | Types of mutations included | Minimum DNA amount (ng) | Known pathogenic variant removal | Germline variant removal approach |
|---|---|---|---|---|---|
| WES (Gold Standard) | 22,000 genes 30 Mb | Somatic, missense mutations and INDELS | 150–200 | No | Matching normal tissue |
| ACTOnco+ | 440 genes 1.12 Mb | Non-synonymous and synonymous | 40 | Yes | Algorithm-based |
| AZ650 | 649 genes 1.65 Mb | Non-synonymous and synonymous | 100 | No | Matching normal tissue |
| OncoPanel v3.1 | 447 genes 1.94 Mb | Non-synonymous only | 50 | No | Algorithm-based |
| SureSelectXT | 592 genes 1.40 Mb | Non-synonymous only | 50 | No | Algorithm-based |
| FoundationOne CDx | 324 genes 0.80 Mb | Non-synonymous and synonymous | 50 | Yes | Algorithm-based |
| TruSight Oncology (TSO500) | 523 genes 1.33 Mb | Non-synonymous and synonymous | 40 | Yes | Algorithm-based |
| JHOP2 | 432 genes 1.14 Mb | Non-synonymous and synonymous | 50 | Yes | Algorithm-based |
| GuardantOMNI | 500 genes 1 Mb | Non- synonymous and synonymous | 40 | NA | Algorithm-based |
| MSK-IMPACT | 468 genes 1.14 Mb | Non-synonymous only | 15 | No | Matching normal tissue |
| NeoTYPE Discovery Profile for Solid Tumours | 372 genes 1.10 Mb | Non- synonymous and synonymous | 20 | No | Algorithm-based |
| Ion AmpliSeq Comprehensive Cancer Panel | 409 genes 1.17 Mb | Non- synonymous only | 30 | No | Algorithm-based |
| PGDx elio tissue complete | 507 genes 1.33 Mb | Non- synonymous and synonymous | 50 | Yes | Algorithm-based |
| QIAseq TMB panel | 486 genes 1.33 Mb | Non-synonymous only | 40 | No | Algorithm-based |
| Oncomine Comprehensive Assay Plus (OCA Plus) | 517 genes 1.06 Mb | Non-synonymous only | 20 | No | Algorithm-based |
| Oncomine Tumour Mutation Load Assay | 409 genes 1.20 Mb | Non-synonymous only | 20 | No | Algorithm-based |

method for TMB assessment and has been extensively used in clinical trials that demonstrated an association between TMB response and the clinical efficacy of ICI treatment. Since it can provide a more accurate and comprehensive estimation of TMB due to its higher sequencing depth and broader coverage of the exome, capturing a broad range of variants, including SNVs, indels, as well as CNVs. However, its incorporation into routine clinical settings is challenging and rather reserved for research purposes, as it requires complex analysis and matching with a normal DNA sample to eliminate germline variants, thus accounting for the somatic genetic aberrations only, and may lead to potential false-negative results in poorly covered regions. Therefore, it is still associated with long turnaround time, high operational costs, and complex bioinformatics for data analysis and interpretation (Abbasi et al. 2021; Pei et al. 2023). Targeted Gene Panel Workflow is also considered a potentially acceptable and reliable way for TMB estimation in clinical practice since it focuses only on a specific subset of cancer-related genes that are known to be more relevant to the tumour biology, allowing for deeper sequencing and higher coverage, and therefore, it's more cost-effective than WES. Thus, large targeted gene panels have been routinely utilised in the clinical settings, and several commercially available targeted gene panels can be used for the TMB quantification. On the contrary, it may potentially miss variants in non-targeted regions, and leading to an underestimation of TMB. Moreover, gene panels vary in terms of the input sample needed, the number of genes and the genes included, the regions covered, the methodology, and the bioinformatics methods. These factors may contribute to discrepancies in the estimation of TMB and, ultimately, its predictive value (Frampton et al. 2013; Chalmers et al. 2017; Allgäuer et al. 2018; FDA unveils a streamlined path for the authorization of tumor profiling tests alongside its latest product action 2018; Meléndez et al. 2018; Büttner et al. 2019; Stenzinger et al. 2020; Meri-Abad et al. 2023; Zhang et al. 2024). Therefore, concordance studies are required to provide a standardised framework, to harmonise data between various gene panels, and translate TMB data from WES into gene panels.

### Somatic mutations and variant calling

Variant calling is also a significant variable in determining the TMB. Various bioinformatics methods or filters are employed to include or exclude certain genetic variants from the TMB assessment. Moreover, there are different types of mutations considered for TMB estimation, such as single nucleotide variants (SNVs) consisting of both synonymous and nonsynonymous mutations, as well as small insertions and deletions (indels). These factors are vital and should be taken into account as they have a direct and significant impact on TMB results (Singh et al. 2013; Koeppel et al. 2017; Hellmann et al. 2018; Sung et al. 2022). WES and NGS gene panels mainly detect SNVs in tumours, thus limiting estimation of TMB and the neoantigen repertoire to missense and nonsense mutations. Although recent studies have demonstrated that responses to immunotherapy are more closely associated with nonsynonymous than synonymous mutations, TMB estimation often does not distinguish between these types of mutations, only the number of SNVs.

There are several steps involved in the calculation of targeted panel-based TMB; first, variant calling and defining the true variants based on quality metrics then the annotation of variant types included for TMB estimation. Second, the filtration of germline mutations and single-nucleotide polymorphisms

(SNPs) to be excluded from TMB calculation. Third, the deployment of an algorithmic adjustment to reduce or eliminate the bias of cancer hotspot mutations. Finally, the use of regression model to validate the TMB estimation methodology (Lauss et al. 2017).

The variant allele frequency (VAF) threshold also varies across NGS panels and TMB platforms. While WES captures variants with VAFs of 5–10%, FoundationOne CDx and Oncomine assays detect variants with a VAF of ≥5% and MSK-IMPACT panel detects hotspot mutations with a VAF of ≥2% and non-hotspot mutations with a VAF of ≥5% (Srinivasan et al. 2002; Jennings et al. 2017; Riaz et al. 2017; FDA 2018b; ThermoFisher Oncomine™ tumor mutation load assay user guide 2018). Moreover, errors in TMB estimation occur due to formalin fixation of samples. DNA damage, artefacts, or sample contamination may all contribute to the overall TMB estimation. To overcome this issue and enhance variant calling, sequencing of both DNA strands is advised (Rizvi et al. 2015; Snyder et al. 2014). Furthermore, TMB estimation becomes complex in terms of its measurement units (mut/Mb versus total mutations/tumour) while comparing the TMB across various studies.

### TMB thresholds for diverse tumour types

TMB is a continuous and even dynamic variable. Differences ranging from 0.001/Mb to >1,000/Mb have been observed across various cancers and even within the same cancer type. Cancers developing in response to chronic exposure to carcinogens, such as melanoma, UV light, and lung cancer to tobacco, exhibit some of the highest TMBs. In contrast, TMB has been found to be low in paediatric, gastrointestinal, and haematological malignancies, whereas breast, kidney, and gynecologic cancers exhibit intermediate TMB levels. The TMB variation is observed not only across different tumour types, but also across different histological subtypes within the same cancer type. For example, lung, head, and neck cancers exhibit less variation in TMB, whereas colon, urothelial, and endometrial cancers show greater TMB heterogeneity (Alexandrov et al. 2013; Chalmers et al. 2017; Zehir et al. 2017; Vanderwalde et al. 2018; Merino et al. 2020). The difference in the prevalence landscape of TMB across various cancer types is shown in Figure 3.

The initial TMB quantification was based on a retrospective exploratory analysis of randomised ICB trials, which used numeric cutoffs of either 178 muts/exome (WES assessment) or 10–20 mut/Mb (targeted gene panels) (Mellman et al. 2011; Fabrizio et al. 2018; Gandara et al. 2018; Hellmann et al. 2018; Ramalingam et al. 2018; Szustakowski et al. 2018). Meanwhile, the most extensively studied and clinically validated approach (prospectively) was used for NSCLC, in the clinical trials of checkmate-568 and checkmate-227, in which the TMB threshold of ≥10 mut/Mb estimated by FoundationOne CDx was established. The determination of a universal TMB threshold that can be used across various cancer types is unlikely, owing to the significant variation in the median number of somatic mutations across tumour types (Blank et al. 2016; Chen et al. 2017; Goodman et al. 2017; Galanina et al. 2018). Thus, further research is required to accurately determine the clinically validated TMB thresholds for each cancer type.

### Some TMB-L tumours respond to ICIs

Another confounding issue is that, although TMB-H has been correlated with vulnerability to ICI therapy, some patients with

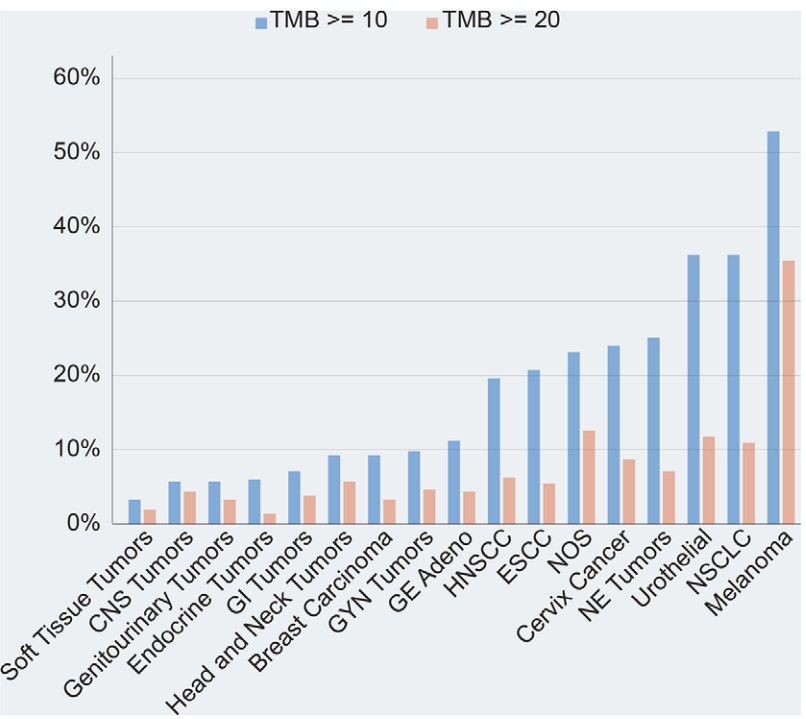

**Figure 3.** Landscape of TMB prevalence across various cancer types.

TMB-L respond to ICIs (Turajlic et al. 2017). For instance, many patients with Kaposi sarcoma achieved complete or partial responses when treated with PD-1 antibodies despite a low TMB (Saeterdal et al. 2001). This result raises questions regarding the role of TMB as a biomarker for the selection of patients who receive immunotherapy, and several questions remain unanswered confounding these results. First, how confident are we in the false negativity of TMB, the heterogeneity across various NGS panels, and the vast technical requirements to accurately run the TMB tests? Second, patients who respond to ICIs often have tumours with a large number of tumour infiltrating lymphocytes (TILs). Thus, the biopsy specimens available for such samples might contain an insufficient proportion of tumour cells relative to TILs, thereby leading to a false-negative or inaccurate TMB status. Finally, some of these studies relied on archival tissues, which might not be representative of the actual genetic status of these patients at the time of treatment.

Furthermore, gene alterations affecting other molecules in the immune response pathway may obscure the significance of the TMB estimation. For example, a recent study demonstrated that tumours with loss of heterozygosity for HLA (HLA-LOH) exhibit higher TMBs compared with tumours without HLA-LOH. However, the downregulation of HLA genes is an immune evasion strategy for cancer cells. Loss of heterozygosity (LOH) in the HLA-I alleles, a total of six different HLA-I alleles at three loci, HLA-A, HLA-B, and HLA-C, is observed in various cancers and has been associated with poor outcome in response to ICIs. A computational tool was recently developed enabling the quantification of the allele-specific copy number of the HLA locus. These algorithms have been shown to help better classify patients into TMB-H and TMB-L groups, and it was found that the HLA-corrected TMB has better predictive power for PFS and OS (McGranahan et al. 2017; Shim et al. 2020). Thus, HLA-corrected

TMB can also help to better predict patients with TMB-H who will not respond to ICIs.

## Conclusion and future directions

There is robust evidence supporting the predictive utility of TMB as a biomarker for response to ICI therapies. Nevertheless, the application of TMB in routine clinical practice remains constrained, while PD-L1 expression continues to prevail as the gold standard for predicting the response to cancer immunotherapy.

This evidence led to the US FDA's approval of tissue-agnostic accelerated approval for pembrolizumab in TMB ≥10 mutations/Mb solid tumours (FDA 2020). However, there are still several unresolved challenges that need to be addressed before considering TMB as a reliable clinical biomarker. The tumour heterogeneity is another concern that can lead false TMB results. This challenge can be addressed by obtaining multiregion sampling and conducting single-cell sequencing in order to overcome the tumours heterogeneity. These approaches aim to provide a more accurate estimation of TMB by capturing a broader spectrum of mutations present within the tumour. Additionally, although the data suggest that TMB is associated with tumour response, > 50% of TMB-H tumours do not respond to ICIs, while around 5% of TMB-L tumours do respond. The fact that some TMB-L tumours such as Kaposi sarcoma respond indicates that additional factors may contribute to ICI efficacy. Thus, TMB alone is not the determining factor in the response to immune checkpoint inhibitors, which raises important questions about how to optimally select patients for ICI treatment and how to overcome the limited ORR of only ~30–50%. Additionally, the biology of tumour immunity is complex and involving various factors beyond genetics. TMB and genomic variants are only a single piece of the tumour immunobiology puzzle. Additional aspects need to be also

investigated and taken into consideration. Immune profiling and fitness: The presence and activity of immune cells within the tumour microenvironment (TME) play crucial roles in modulating the antitumour immune response. Therefore, additional biomarkers such as tumour-infiltrating lymphocytes and immune gene expression profiles can provide relevant information about the TME. Tumour-specific antigens: TMB focuses on the total number of mutations in the tumour genome, but not all mutations generate immunogenic neoantigens that can elicit an effective immune response; thus, biomarkers that identify the presence and recognition of tumour-specific antigens, such as neoantigen burden or HLA expression, can provide relevant insights into the potential immunogenicity of the tumour (Apavaloaei et al. 2020; Bubie et al. 2020). Although the appeal of TMB as a marker is that knowledge of the exact mutations may not be necessary, just the number of them, the specific mutations revealed in WES or NGS panel analysis may also be exploited for other treatment options. For example, the mutations revealed in the TMB analysis could be subjected to further analysis for the best 8 to 10 candidates for MHC presentation. Such prediction algorithms exist, and in combination with the technology of mRNA vaccines, may be an alternative method to use the somatic mutations in human cancer in combination with ICI treatment (Sahin et al. 2017). TMB focuses only on small somatic mutations; however, other genomic alterations such as gene amplifications, fusion, and rearrangements, may also impact tumour immune responses. Thus, the integration of these alterations can provide a more holistic understanding of the tumour immune landscape.

Another important factor to be considered is the variability and limitations of the sequencing methodology or workflow, either WGS, WES, and targeted panels which significantly impact the TMB quantification. The current technology and analysis of WES render it impractical for its routine implementation in clinical practice. It is imperative to devise a harmonised/standardised approach for various targeted gene panels to ensure the accuracy of TMB quantification. Owing to TMB inter-variability between cancers, it is critical to determine tumour-specific and optimal TMB cutoff points. Instead of the current classification of high or low TMB, a novel three-tier TMB scheme (low, intermediate, and high) was proposed to reduce TMB misclassification. Several academic and commercial laboratories have participated in the Friends of Cancer research TMB harmonisation to ensure consistency across panels and have come up with the following recommendations and best practices (Vega et al. 2021):

1. The analytical validation of the various NGS panels should follow a standard and aligned path to ensure the sensitivity and reliability of TMB values, irrespective of the type of panel or bioinformatics pipeline used.
2. The consortium recommends consistency in reporting TMB results as (mut/Mb) to keep TMB values comparable and interpretable across different platforms.
3. Alignment of TMB thresholds using a calibration curve that compares and validates data across different panels is recommended.

Once the standardisation of cross-NGS assays has been completed, it is imperative that TMB be tested in larger prospective clinical trials with a preplanned endpoint and a clear TMB threshold to validate and consolidate the predictive efficiency of TMB as a biomarker of response to immunotherapy and to determine the best ICI therapy. It should also be determined whether TMB can be used on its own as a single variable or in combination with other biomarkers. This raises an important question about how

better strategies to optimally identify responders for ICIs treatment and/or exclude those are unlikely to achieve responses and avoid the unnecessary AEs. One strategy could be combining TMB with other biomarker(s) or developing a mutational and/or immunogenic score to better select patients for immunotherapy intervention.

Finally, the advances in liquid biopsy or circulating tumour DNA biopsy can play an important role in overcoming issues related to tissue availability and invasiveness of the biopsy surgical procedure. The estimation of TMB using blood samples makes it possible to assess bTMB at any time before or during treatment, can also overcome the DNA quality during the fixation process as well as the spatial and temporal heterogeneity of the tumour. The implementation and utility of bTMB have been successful in several trials, including POPLAR and OAK for atezolizumab, and MYSTIC for durvalumab and tremelimumab.

**Open peer review.** To view the open peer review materials for this article, please visit http://doi.org/10.1017/pcm.2024.6.

**Acknowledgements.** This review is a part of the dissertation of the MSc in Precision Cancer Medicine at the University of Oxford, and thus I wish to acknowledge and thank the program director, Professor Anna Schuh, for the guidance and educational support throughout the development of the concept and writing of the dissertation. I would like to acknowledge and thank Helene Dreau, MSc, and Tracy Bye, PhD, for their great assistance, as well as the library manager at the Bodelian Libraries, Carolyn Smith, for her support and guidance in navigating several datasets and the proper use of reference manager. Moreover, I would like to thank Nataliia Kiptenko for her great assistance in the graphical and medical illustrations.

**Author contribution.** A.M.E: conceptualisation, methodology, project administration, writing; original draft preparation, reviewing and editing.

**Financial support.** This work received no external funding.

**Competing interest.** The author declare that the research was conducted in the absence of any commercial or financial relationships that could be construed as potential conflicts of interest.

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
