## [Reviewer Report]

This manuscript describes applying analysis of Tumour Mutational Burden to clinical decisions on immunotherapy. The technical approach, pitfalls, and need for standardization and further validation are well covered. Some points to address include:

1. “Several inhibitory immune checkpoints, such as CTLA-4, PD-1, LAG-3, TIM-3, and TIGIT have been identified.” Add “as therapeutic targets for immunotherapy.”

2. “Tumors with high mutational load, such as melanoma and non-small cell

lung cancer (NSCLC), responded to ICI therapy, and thus gave birth to tumor mutation burden (TMB) as a potential biomarker.”

Might say that melanoma was historically expected to respond to immunotherapy but responses in lung cancer were unexpected (cite Brahmer paper) and since lung cancer was known to be highly mutated, focused early investigators on the relationship between TMB and response.

3. “TMB is defined as the total number of somatic mutations within the tumour genome.” This is not accurate; many of the assays look at just a subset of the genome. Perhaps say “TMB is rigorously defined as the total number of somatic mutations within the tumour genome but in practice involves an estimate from a subset of the genome.”

4. “Gene panels of ≥ 0.8 Mb are therefore essential for the precise TMB” Change “precise” to something like useful or accurate

5. Explain in the text the “pathogenic variant removal” in Table.

6. Kaposi’s sarcoma is an example of a virus-caused tumor (human herpesvirus 8). This and other virus-caused tumors have antigens encoded by the virus which are not captured by TMB. Discuss immune recognition of tumor virus antigens.

7. asking for “devise a harmonised/standardised approach” with so many assays seems unrealistic. Be explicit about what feature(s) would have priority for standardization.

8. “analytical validation of different NGS panels”, explain what analysis needs to be done in more detail.

9. “better strategies to optimally select patients for ICIs treatment and overcome the limited ORR, only 30%–50%.” This statement reads as if the assay is going to increase the number of responders. I don’t believe better selection will increase the number of responders but by excluding those unlikely to respond, will increase the percentage. This may be a more efficient use of resources and may direct those unlikely to respond to alternative treatments or trials. Revise.

10. Should mention the studies that show patients with Mismatch repair deficient have high TMB and responses, FDA approval. In clinical practice, how many MMRD patients are likely to be identified by MMRD assays versus TMB assays.

11. In the table, where germline variant removal is “Algorithm based”, explain in the text what is done.

12. The authors say how many ng of DNA are needed for a TMB analysis in the Table but please provide context by saying how much tissue is needed; 100 mg cells or one slide or whatever?

---

## [Reviewer Report]

Review of Elbehi et al.

The Challenges and Opportunities of Applying Tumour Mutational Burden Analysis to Precision Cancer Medicine

This review summarizes some of the opportunities and challenges of applying TMB to precision medicine. There are a number of inaccuracies in this review and inappropriate citations, which needs to be corrected. The discussion can be improved by making them more completed. Some of the main sections are very underdeveloped. Specific comments below.

1. Page 3

“TMB is defined as the total number of somatic mutations within the tumour genome.”

In fact, there are a number of definitions for TMB and different ways of measuring it. The author should be more nuanced in describing.

2. Page 3….“Further, higher TMB corresponds to a higher number of somatic mutations and high neoantigen load. Thus, there is an increasing probability that these neoantigens could be recognised d by cytotoxic T-cells and elicit an immunogenic response, leading to destruction of cancer cells. 14-17”

References 14-17 are not the appropriate references for supporting the statement. Most definitely not the Ott et al vaccine trial. It is important to cite properly. The first reports that high TMB as well as DNA damage repair mutations associate with good response to ICIs are PMID: 25765070, PMID: 25409260. Separate citations for the ability of T cells to target neoantigens should also be included here:

Mandelboim O., Berke G., Fridkin M., et al. 1994. CTL induction by a tumour-associated antigen octapeptide derived from a murine lung carcinoma. Nature 369:67.

Mandelboim O., Vadai E., Fridkin M., et al. 1995. Regression of established murine carcinoma metastases following vaccination with tumour-associated antigen peptides. Nat. Med. 1:1179.

Robbins P. F., El-Gamil M., Li Y. F., et al. 1996. A mutated beta-catenin gene encodes a melanoma-specific antigen recognized by tumor infiltrating lymphocytes.

3. Ref 18 is not the primary reference(s) reported for TMB in lung cancer. The author needs to make sure reference citation is accurate

4. Studies on bottom of page 3 and top of page 4 not cited.

5. “Based on these results, the US FDA granted an accelerated approval to pembrolizumab for the treatment of adult and pediatric patients with unresectable or metastatic TMB-H (≥ 10 mut/Mb) solid tumours that progressed after prior treatment and had no satisfactory alternative treatment options. 19-21

Reference 19 and 20 not appropriate. Van allen study doesn’t even deal with anti-PD1. Need to correct inaccurate citations. Please pay attention to what studies support statements.

6. Data review in section 2 is very incomplete. Many critical supportive studies not discussed.

7. Discussion of Variation of sequencing quality is superficial. It would be helpful to address advantages and disadvantages of each depending on workflow.

8. Pg. 5. Rather than just saying that depth is important, it might be helpful to include real observations on how sensitivity and specificity can change as a function of depth and quality of sequencing. All this has been published and would be useful to mention here.

9. The summary in part C is very superficial.

10. Section G. TMB is one variable to consider. Rather than ask open ended questions, the author should compare these to what affects other biomarkers. Sample heterogeneity is a problem affecting all tissue sample based assay and yet this wider context was not mentioned. Moreover, genetics is only a portion of the biology of tumor immunity and hence, TMB should be used in concert with other biomarkers. This was also not explored.

In summary, this review provides a brief summary of the TMB field. However, there are many erroneous citations and many sections need to be developed further for this review to be useful for the readership.

---

## [Editor Report]

Dear Dr Elbehi, Thank you for your submission to Cambridge Prisms: Precision Medicine. The subject topic of tumour mutational burden (TMB) and associated analysis is timely and a comprehensive and up-to-date article on the topic welcome. We had two peer reviewers provide comprehensive review of submitted paper and from these, we advise that major revision is needed in order for the manuscript to be considered for publication. Reviewer 1 provides helpful and specific examples of where technical clarification should be provided. However, please note that Reviewer 2 highlights topic sections which have been insufficiently developed in detail and/or where incorrect referencing has been provided which requires correction and inclusion of further appropriate discussion. Please kindly address these issues as highlighted in order for the progress of this paper.

---

## [Editor Report]

The authors have diligently addressed most of the issues raised by the reviewers, demonstrating their commitment to improving the article. However, further enhancements are still needed.

Major comments:

The authors should clearly state that the role of TMB in routine clinical settings is currently limited (there is only one FDA-approved indication), and the principal method for patient selection PD-1/PD-L1 checkpoint inhibitors remains PD-L1 expression.

Liquid biopsy (ctDNA) has emerged as an attractive alternative or complementary method to tissue-based TMB testing. There is extensive literature on this approach, including its use in the clinic (e.g., see a recent review at https://www.ncbi.nlm.nih.gov/pmc/articles/PMC9853269/). Currently, there is only a short remark on this topic in the Conclusion and Future Direction section. The Authors should give it more attention (e.g., specify the blood material, mention the use of bTMB in detecting and monitoring minimal residual disease), and add a few references.

Minor comments

In the sentence: “have been identified as therapeutic targets for immunotherapy and some of them have shown positive results in randomized phase 3 trials and been approved such as PD-1, LAG-3 & CTLA-4 remove the part: ”and been approved such as PD-1, LAG-3 & CTLA-4", as this is repeated in the following sequence.

The statement, “… due to the high number of mutations in lung cancer, the relationship between TMB and treatment responses has been extensively investigated,” is imprecise. Actually, the high number of mutations in lung cancer prompted immunotherapy approaches in this malignancy, not TMB testing.

In the sentence: "This evidence led to the US FDA’s approval of tissue-agnostic accelerated approval for pembrolizumab in TMB ≥10 mutations/Mb solid tumors (FDA, 2020)", provide reference to this specific approval.

There are some grammatical and spelling errors which need to be corrected.

---

## [Editor Report]

The authors have correctly addressed earlier issues. There are only two suggestions for phrasing two paragraphs that they might wish to consider.

Page 3, lines 34-46

Finally, the field of analyzing circulating tumor DNA (ctDNA), known as liquid biopsy, is advancing very rapidly. It holds significant potential to overcome many of the above-mentioned challenges. The sequencing of ctDNA can provide important insights into the evolution of the cancer mutational landscape. It can also act as a real-time biomarker for the timely and accurate estimation of TMB. Moreover, liquid biopsy can provide a non-invasive tool for the real-time monitoring of responses to therapy, assessment of minimal residual disease and the detection of early sign of disease progression. 30

Page 10, lines 27-34

There is strong evidence supporting the predictive value of TMB as a biomarker for response to ICI therapy. Despite this, the utilization of TMB and its incorporation into routine clinical settings is very limited, and PD-L1 expression continues to be the gold standard for predicting the response to immune checkpoint inhibitors. Indeed, only one tissue-agnostic accelerated the US FDA’s approval for pembrolizumab in TMB ≥10 mutations/Mb solid tumors (FDA, 2020). There are still….

Use the British English spelling consistently.